# Microbiome and Metabolomics in Liver Cancer: Scientific Technology

**DOI:** 10.3390/ijms24010537

**Published:** 2022-12-28

**Authors:** Raja Ganesan, Sang Jun Yoon, Ki Tae Suk

**Affiliations:** Institute for Liver and Digestive Diseases, College of Medicine, Hallym University, Chuncheon 24253, Republic of Korea

**Keywords:** liver cancer, microbiome, metabolomics, metabolites, scientific technology

## Abstract

Primary liver cancer is a heterogeneous disease. Liver cancer metabolism includes both the reprogramming of intracellular metabolism to enable cancer cells to proliferate inappropriately and adapt to the tumor microenvironment and fluctuations in regular tissue metabolism. Currently, metabolomics and metabolite profiling in liver cirrhosis, liver cancer, and hepatocellular carcinoma (HCC) have been in the spotlight in terms of cancer diagnosis, monitoring, and therapy. Metabolomics is the global analysis of small molecules, chemicals, and metabolites. Metabolomics technologies can provide critical information about the liver cancer state. Here, we review how liver cirrhosis, liver cancer, and HCC therapies interact with metabolism at the cellular and systemic levels. An overview of liver metabolomics is provided, with a focus on currently available technologies and how they have been used in clinical and translational research. We also list scalable methods, including chemometrics, followed by pathway processing in liver cancer. We conclude that important drivers of metabolomics science and scientific technologies are novel therapeutic tools and liver cancer biomarker analysis.

## 1. Introduction

Hepatocellular carcinoma (HCC) is the second most common cancer-related cause of death worldwide, and, both domestically and internationally, incidence rates are rising. HCC is globally caused by two conditions: alcoholic liver disease (ALD) and non-alcoholic fatty liver disease (NAFLD) [1,2]. Every year, approximately 750,000 novel instances of liver cancer are recorded worldwide. According to population-based interventions, the liver cancer proliferation rate continues to be close to death, meaning that the majority of patients who develop HCC die from it. According to data, the five-year survival rates in the United States have increased slightly to approximately 26%. This expansion is thought to be a result of better surveillance in high-risk patients who can be identified (those who have hepatitis B and C viruses), as well as clinical intervention (resection or transplant) for patients with early-stage disease [3].

Guidelines have been published by a number of organizations, including the National Comprehensive Cancer Network (NCCN), the European Association for the Study of the Liver (EASL), and the American Association for the Study of Liver Disease (AASLD), to normalize the approaches to judgment and treatment [4,5,6]. The earlier that HCC is detected and treated, the better the prognosis, as is true for the majority of disease processes. The observation of patients who are known to be at a high risk provides the best opportunity for an early diagnosis. Both people who have cirrhosis from any cause and hepatitis B carriers fall under this category [5]. According to the 2012 NCCN guidelines, high-risk patients should receive liver ultrasonography and AFP screenings every six to twelve months. A hepatic nodule larger than 1 cm accompanied by an increasing AFP should be evaluated.

Over the past ten years, the criteria for the diagnosis of HCC have changed. The AASLD, NCCN, and EASL working groups have developed imaging criteria that effectively predict malignancy so as to reduce the necessity for a percutaneous biopsy and its accompanying risks in patients with underlying liver conditions (tract seeding, hemorrhage, etc.) [4,6]. On contrast-enhanced computed tomography (CT) or magnetic resonance imaging (MRI) images, early arterial enhancement and venous phase washout, which are related to the fact that these hypervascular lesions are primarily supplied by branches of the hepatic artery, are imaging markers of HCC. In the context of chronic liver disease, HCC refers to tumors larger than 1 cm in size that have certain imaging characteristics on triple-phase CT or contrast-enhanced MRI.

*Escherichia, Pseudomonas, Lactobacillus* and other gut bacteria are crucial to the ‘gut origin of sepsis’ theory. The dominant signs of the gut microbial imbalance are significant increases in gram-negative bacteria such as *Escherichia coli* and the *Atopobium cluster,* which includes the genera *Atopobium, Coriobacterium, Collinsella,* and *Eggerthella,* as well as significant decreases in *Bifidobacterium, Enterococcus,* and *Lactobacillus* species [7]. Moreover, it was discovered that *Fusobacterium nucleatum* was prevalent and abundant in patients with cancer [8].

The previous recommendations required typical enhancement on both imaging modalities (CT and MRI) for lesions between 1 cm and 2 cm to define HCC. Although the imaging standards have changed, only lesions larger than 2 cm and exhibiting typical enhancement qualify as Model for End-Stage Liver Disease (MELD) exemption points for liver transplantation. To more accurately characterize lesions that do not fulfill these criteria on standard arterial and venous phase imaging alone, some facilities have utilized MRI with new contrast agents, such as gadoxetic acid. On T1-weighted (hepatocyte phase) imaging, lesions that could be HCC are darker than the surrounding liver [9]. Despite indications of a better diagnosis accuracy, gadoxetic acid-enhanced MR imaging has not yet altered the paradigm used to determine therapy eligibility. Thus, despite an increased imaging specificity, gadoxetic acid-enhanced MR imaging has not yet altered the diagnostic pattern used to establish clinical prevention [10].

## 2. Enabling Technologies for Metabolomics Research and Engineering

Microbiome-derived metabolomics or metabolomics profiling refers to the detection of metabolites or small molecules in gut microbial communities that are related to alcoholic liver disease (ALD) and non-alcoholic fatty liver disease (NAFLD) [11,12]. Metabolomics science, a targeted and untargeted profiling method, involves the large-scale study of the metabolic complements of the cells and has the ability to provide adequate coverage of the metabolome. Accurate quantitative information can be provided with wide-spanning technical care for its use in the analysis of metabolic oscillations in the gut microbial environment [13,14]. Metabolomics is a promising platform for the identification of potential responses to stimuli, molecular signatures, and organic compounds that are closely related to metabolic phenotype and therapeutic biomarker discoveries [15,16]. The isotopes of ^1^H-, ^13^C-, ^14^N-, ^19^F-, ^31^P-, and ^43^Ca-rich metabolites in liver cells have led to the development of therapeutic screening applications [11,17].

The metabolomics profiling of microbial metabolites and their computational technologies act as a high-throughput global analytical platform. Metabolomics can illustrate small molecules (molecular weight < 1 kDa) [18,19,20]. Figure 1A shows the long history of metabolomics. Metabolites, or small molecules, are the fundamental output of combined microbiome and host interactions that may provide signatures of gut-microbiome-mediated ALD conditions.

In Figure 1B, and Figure 1C, the guiding principles of genomics, transcriptomics, proteomics, and metabolomics are listed with targeted and untargeted profiling methods, each with their own benefits and limitations [21,22,23]. Untargeted metabolomics is focused on the examination of recognizable metabolites and/or metabolomes in biological mixtures, including unknown chemicals. The metabolome is the set of metabolites within a given cell. Metabolome concentrations are widely connected with phenotypic expression [17,24].

In omics sciences, gut-microbiota-based liver therapeutic candidate metabolome screening and metabolomic profiling are significant. Technically, metabolomics has already entered the clinic, with applications in various liver disease screenings. Many metabolomic-signature-based clinical tests can be used to quantitatively analyze low-molecular-weight metabolites in cells, tissue, and/or biofluids [11,25]. Metabolomic signatures have been connected to phenotype expression, which acts as a functional endpoint of a biochemical reaction. The metabolome is the quantification of metabolites that result from the interplay between many domains [26,27,28]. Microbial metabolomic signature-based liver cancer represents the most ‘cutting-edge’ example of metabolomics, enabling precision medicine.

Mass spectrometry (MS)-based data analysis and peak identification have been used to explore the regulation of the biological actions of the gut microbiota and host–microbiome relationships by utilizing metabolomics methods [29,30]. The quantitative analysis of microbiome-derived small molecules delivers a functional read-out of cells. MS is a more prominent platform in metabolomics than nuclear magnetic resonance (NMR). MS has become more extensive in host-microbiota analysis because of its high sensitivity, high-throughput discovery, and wide variety of metabolome analyses [31,32,33,34]. While NMR can evaluate metabolites in the micromolar range, the utilization of MS licenses the discovery of up to nanomolar concentrations. MS is likewise effectively connected with chromatographic partition, decreasing the impacts of biological samples as well as restricting the complexity of analytes at the time of identification [35,36,37,38].

Gas/liquid chromatography (GC/LC) has become the most applied chromatography-MS device for the investigation of both polar and nonpolar metabolites [39,40]. GC/LC-MS has been applied for the examination of various volatile and nonvolatile compounds and for important metabolites after derivatization. Capillary electrophoresis (CE)-MS (CE-MS) is also used for the examination of polar, charged metabolites, as explored in the previous literature [41,42]. LC/GC-MS focuses on changes in mass-to-charge (*m*/*z*), with NMR spectroscopy providing the spectral intensities. The analytical characteristics of NMR, MS, Raman micro spectroscopy, immunochemistry, and enzymatic assays are briefly discussed in Table 1. Currently, the NMR and LC/GC-MS methods offer high-quality metabolomic datasets.

In metabolomics investigations, NMR operates with a lower sensitivity than MS-based methods. NMR spectroscopy can quantify and target the metabolites in biofluids, with rapid sample preparation. When the sample complexity can be mitigated, NMR delivers valuable structural information. NMR is valuable for identifying gut microbiota-derived compounds (i.e., amino acids, lipids, fatty acids, organic and inorganic metabolites) [43,44,45,46]. However, both MS and NMR allow for small-molecule profiling and the identification of the high diversity of microbial products.
ijms-24-00537-t001_Table 1Table 1Application, analytes, detection, and comparison of top analytical devices for metabolomics scientific technology.AnalyticalDevicesScientificInstrumentsKey FunctionsMetabolic ApplicationsRefMS
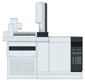

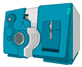
High throughput, high sensitivity/resolution, capable of quickly examining, reduces complexity,improves resolution, specificity, and quantification, allows for isotopic labeling, offers structural information, performed under ambient environmental conditions with the preservation of tissue morphology (DESI-MS).GC-MS: SCFAs and ketones, carbohydrate metabolites, amino acids. LC-MS: amino acids and their byproducts, bile acids, lipids and fatty acids, sugar metabolites, vitamins, and related compounds.Imaging MS: MALDI/DESI-IMS and nano SIMS. [31,32,33,34,36,37,38,43,47].NMR
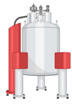
Affords structural evidence, low-sensitivity compared to MS, high-throughput, permits the quantification of isotopic labeling, delivers spatial data (NMR imaging or MRI).Sugar metabolites, amino acids and amino acid byproducts SCFAs, vitamins, untargeted analysis, and metabolome finger printing.[22,23,44,45,48] RamanMS
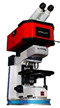
3D info, high-throughput, structural data, non-destructive methods, lower sensitivity versus MS and NMR.This can be united with fluorescent probes and isotopic labeling for the single-cell-resolved assessment of nutrient assimilation.[43]UHPLC
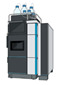
High-sensitivity detection Detection and identification of a broad range of metabolites[49,50]Immunochemistry and enzymatic assays--Low-throughput, high specificity, may provide spatial information (immunohistochemistry or immunofluorescence).Eicosanoids, uric acid, serotonin, neurotransmitters, lipopolysaccharide, some vitamins, sugar metabolites.[51,52] Abbreviations: DESI-MS, desorption electrospray ionization mass spectrometry; GC-MS, gas chromatography–mass spectrometry; LC-MS, liquid chromatography–mass spectrometry; Raman MS, Raman micro spectroscopy; UHPLC, Ultra-High-Performance Liquid Chromatography; MALDI, matrix-assisted laser desorption; nano SIMS, nanoscale secondary ion mass spectrometry; SCFA, short-chain fatty acid; Ref, References.

Both NMR and MS-based metabolomics have been applied to study the gut microbiota via isotope tracing in nutrient accommodation. Metabolic alterations using isotope labeling remain challenging because of their structural exchange in hosts and various microorganisms and the difficulty of identifying the paths of small molecules and/or metabolites [43,44]. The topographies of MS, nanoscale secondary ion MS, and Raman spectroscopy deliver high-throughput three-dimensional data, which are shared with fluorescent probes and stable isotope tracing to achieve a single-cell resolution within host and gut microbial cells [43,44].

Targeted metabolomics profiling and lipidomic profiling have been used to measure defined groups of metabolites. The methodologies can be defined by the quantity of notable metabolites and the reliability of the quantification of a specific approach. Here, reliability is introduced either as the exactness of entire quantifications, normally transferred in micromolar units, or as accuracy, given by semiquantitative judgments in normalized units [11,24,25]. The best-accuracy approach could be hypothetically accomplished when an isotopically considered internal standard of a specific metabolite is spiked in biofluids during extraction at different concentrations (isotope dilution mass spectrometry). A slightly less reliable technique utilizes an alignment curve of a specific standard spiked at various concentrations, standardized to a spiked constant concentration of an internal standard [26,53]. Table 2 shows the computational tools used for NMR- and MS-based metabolomics analysis in biological samples, focusing on the main breakthroughs in this field.

## 3. Diagnostics Test of Liver Cancer

Various types of medical testing can be performed to identify liver cancer tumors. However, the following common diagnostic exams are performed:I.Physical examination: A general practitioner or gastroenterologist can examine the patient to learn about their health history and identify general risk factors for the development of liver cancer. Examinations include those of the skin, eyes, and areas of the abdomen (signs of jaundice). Additional tests could be necessary to identify the cause of symptoms, depending on the results of the initial physical exam [80,81].II.Radiology tests and imaging: As the name suggests, imaging findings use X-rays, magnetic fields (MRI), or sound waves to provide precise visual scans of internal body regions (ultrasound). Other common tests used to assess liver cancer include CT scans, bone scans, and angiography [82,83].III.Laparoscopy: For the improved viewing of the liver tissue and adjacent organs, laparoscopic surgeries use a small tube with a camera introduced into the abdomen. Diagnostic laparoscopy is a minimally invasive, low-risk surgical treatment that calls for tiny incisions [84]. An improved understanding of the liver cancer’s current stage, assistance in developing a personalized stem cell treatment strategy, or confirmation of an earlier diagnosis can all be achieved with laparoscopy [85].IV.Liver biopsy: A surgical procedure called a liver biopsy uses a sample of the patient’s liver tissue to identify the presence of cancer cells [86].V.Lab tests and blood panels: Lab tests and blood panels are relatively affordable and efficient tools for checking the health of the body and internal organs, monitoring the success of therapy, looking for cancer indicators, or checking for cancer recurrence [87,88].VI.Genetic screening for cancer: Circulating tumor DNA (ctDNA) analysis is distinct from previously known conventional diagnostic techniques. Cancer biomarker tests such as ctDNA analysis only need small saliva samples or cheek swabs, as opposed to invasive tissue biopsies [89]. Rapid screening is a reliable method of prognostic marker detection. This method can detect potential metastatic disease very early, monitor treatment, and identify genetic and epigenetic changes resulting from primary tumors [90].

## 4. Microbiome Research and Engineering in HCC Metabolism

Physiological responses in the host are maintained and coordinated by metabolites produced by the microbiota. The liver cancer processes that HCC may impact through bacterial metabolism, such as these metabolic transportation pathways and their efficacy, must be understood [91]. It is important to comprehend and describe the variety of metabolites that the gut microbiota excretes. Metabolomics has been widely used to illustrate the metabolites produced by gut microbes, particularly in relation to the disease states of the host that they may affect. Metabolomics generally no longer needs to be defined, as we have thoroughly examined this technology [92,93,94].

In an early study, Nicholson’s team discovered significant metabolic differences between germ-free mice and their healthy counterparts for so-called ‘cometabolites’ such as hippuric acid, which is produced when benzoic acid and glycine are conjugated [95,96]. The gut bacteria produce benzoic acid by converting chlorogenic acid into quinic acid, which is then aromatized [97]. The liver receives benzoic acid via the portal supply, where it is conjugated in the mitochondria by first forming a CoA intermediate and then adding glycine [98].

Phenylacetylglutamine is another instance of a metabolism in which phenyl acetic acid is formed from phenylalanine by the gut microbiota [36] and conjugated with glutamine in the host’s hepatic mitochondria [99]. It has been documented that the three aromatic amino acids phenylacetic acid, tyrosine, and tryptophan are all converted into nine aromatic acids via the gut symbiont *Clostridium sporogenes* and circulate in human plasma. These compounds, such as tryptophan-derived indol-3-ylacetic acid, are almost certainly co-metabolized by the host [99].

Dietary flavan3-ol polyphenols, epigallocatechin gallate, epigallocatechin, epicatechin, and catechin may also be sources of aromatic acid metabolites by gut microbiota through intricate oxidation and dihydroxylation pathways that ultimately result in phenylacetic acid, benzoic acid, and catechol [100]. An examination of 143 organic acids frequently found in the urine of healthy individuals revealed that a sizable portion of these were created by the host’s microbiota, with some of them being further digested by the host [101]. Figure 2 shows the three-way relationship between the host, the tumor microenvironment, and the microbiota.

An important group of acidic metabolites generated by the gut microbiota is short-chain fatty acids (SCFAs: acetate, propionate, and butyrate), which are created by gut bacteria during the anaerobic fermentation of plant structural polysaccharides such as cellulose or fiber [102]. The lipopolysaccharide (LPS)-producing genera (*Neisseria, Enterobacteriaceae,* and *Veillonella*) were more abundant among liver cancer–HCC, and butyrate-producing genera (*Clostridium, Ruminococcus,* and *Coprococcus*) were less abundant. Three more biomarkers—*Enterococcus*, *Phyllobacterium,* and *Limnobacter*—can also be used to reliably detect liver cancer.

In cellular environments, SCFAs represent a sizable proportion of energy metabolites, as it is estimated that they provide 60–85% of an animal’s energy needs [102]. In HCC patients with dysregulated fatty acid metabolism, the β-oxidation process of fatty acids is associated with a worse prognosis. The proximal colon of a human contains the largest concentration of SCFAs, where they are both absorbed into the bloodstream and utilized locally by enterocytes [103,104].

In mice with altered gut commensal bacteria, the increase in NKT cells and the suppression of liver tumor growth may both be reversed by colonizing bile acid-metabolizing bacteria (*Clostridium scindens*) and supplementing with secondary bile acids (lithocholic acid or muricholic acid).

The G-protein-coupled receptors GPR41 and GPR43 (free fatty acid receptors) and the niacin receptor GPR109A are only a few of the receptors that can recognize SCFAs, which are present in the colon at concentrations of 50–200 mM. SCFAs can regulate gene expression by inhibiting histone deacetylases in this location and act as signaling molecules that are recognized by particular receptors [105].

## 5. Microbiome Metabolism for Therapeutic Applications in HCC

Table 3 has summarized the microbiome in HCC. The third most common cause of cancer-related deaths globally is HCC, which carries a heavy disease burden [106]. HCC is notorious for being highly aggressive and is associated with frequent progression and recurrence. Numerous immune checkpoint inhibitors, mainly anti-PD-1/anti-PD-L1 monoclonal antibodies, have been studied and approved for HCC over the past few years. However, only a small portion of patients (20%) benefit [107,108,109]. To date, there are no known indicators that can accurately predict the clinical outcome of anti-PD-1/anti-PD-L1 immunotherapy [110].

Additional changes to the microbiome occur during the cirrhosis–HCC transition period. The microbial bacteria of *Veillonella, Streptococcus, Clostridium,* and *Prevotella* were more prominent in the cirrhosis patients. *Eubacterium, Alistipes*, and *Faecalibacterium prausnitzii* were comfortable in the healthy gut–liver axis [123]. The probiotics are lactic acid bacteria, including species of *Lactobacillus, Streptococcus,* and *Enterococcus,* as well as yeast, *Bifidobacterium, Propionibacterium, Bacillus, Escherichia coli,* and *Bifidobacterium,* which can both encourage the growth of helpful bacteria and inhibit the growth of harmful bacteria [124]. *Actinobacteria* were shown to be more prominent in early HCC vs. cirrhosis [125]. The correct diagnosis of liver cancer may be achieved by using three biomarkers (*Enterococcus, Phyllobacterium,* and *Limnobacter*). Among HCC patients, the abundances of the genera that produce butyrate (*Clostridium, Ruminococcus,* and *Coprococcus*) decreased, while the abundances of the genera that produce LPS (*Neisseria, Enterobacteriaceae,* and *Veillonella*) increased [121]. Figure 3 explains the principal mechanisms of liver cancer and liver damage and the metabolic alterations involved.

An initial piece of evidence that a gut microbe may influence liver cancer was discovered in mice that had spontaneously acquired *Helicobacter hepaticus (H. hepaticus)* infections. The *H. hepaticus* infection has been linked to chronic hepatitis and fibrosis in male BALB/c mice. The *H. hepaticus* is a spiral bacterium that also lives in the bile *canaliculi,* the cecal mucosa, and the colonic mucosa and causes chronic active hepatitis and liver tumors. The *H. hepaticus* is the prototypical carcinogenic bacteria for mice, and experimental infection has already been employed as a model of microbial tumor promotion in the liver [126,127,128]. Later, it was revealed that *H. hepaticus* intestinal colonization was sufficient to produce aflatoxin B1 (AFB1)- and HCV transgene-induced HCC. This information prompted crucial research. In addition to activating Wnt/-catenin, hepatocyte turnover, and the impaired phagocytic clearance of injured cells, the processes also implicated stimulated NF-B-regulated networks linked to innate and adaptive immunity [129].

## 6. Liver Transplantation for HCC

The best treatment for both malignancies and the underlying liver condition that most cases of HCC emerge from is generally thought to be liver transplantation. The size and number of tumors determine whether a patient is eligible for a transplant, and standards have been set up to improve outcomes for people with particular types of cancer. The Milan criteria [130], which allow patients with up to three foci of HCC that are less than 3 cm in diameter or one tumor that is less than 5 cm in diameter to receive a liver transplant, are the most often utilized standards globally.

The five-year survival percentage for these patients (75%) was comparable to the survival rate seen in transplant patients at the time who were not cancer patients [130]. The University of California at San Francisco (UCSF) has released its guidelines for liver tumor size regarding tumors measuring less than 6.5 cm, which are observed in one to three tumors. The total tumor diameter should not exceed 8 cm in light of outcome-based evidence with less stringent criteria. There is no negative effect on overall survival in liver cancer [131,132].

Downstaging patients into Milan or UCSF criteria has become a viable method of patient selection as a result of advancements in liver-directed therapy for HCC. What has become clear is that malignancies with a high risk of recurrence after a transplant are those with disease progression despite liver-directed therapy. Scenters can choose patients with better biology and increase patient eligibility without compromising cancer-specific survival by requiring proof of a response to liver-directed therapy prior to the transplant in conjunction with long-term surveillance before deciding to undergo a transplant [133].

## 7. Systemic Chemotherapy Drugs and Approaches to Improving HCC

The US Food and Drug Administration has approved sorafenib, sunitinib, brivanib, linifanib, sorafenib plus erlotinib, vandetanib, nintedanib, dovitinib, and sorafenib plus doxorubicin for treating HCC. These chemotheraputic drugs act as first-line chemotherapy. Numerous chemotherapeutic drugs have been examined as first-line treatments for patients with advanced HCC. Since its approval, the number of HCC patients receiving treatment with the medicine has increased significantly, irrespective of their tumor stage. Phase 2 findings in patients with advanced metastatic HCC support the use of sorafenib, with the treated group demonstrating a nearly three-month survival advantage over the untreated group. The phase 2 chemotheraputic drugs of brivanib, everolimus, S-1, axitinib, GC33, and tigatuzumab act as valuble drugs for liver cancer. The objective response rate is currently at around 2%, with the majority of the benefit being attributed to the stable disease rate, which was observed in phase 2 and 3 trials to range from 35% to 71%. In the phase 3 study, more than 80% of the participants had previously undergone liver-directed therapy (chemoembolization). Phase 3 chemotheraputic drugs such as lenvatinib, sorafenib plus resminostat, regorafenib, cabozantinib, ramucirumab, and tivantinib (ARQ 197) are promising drugs for HCC [134,135,136].

Sorafenib is a currently available therapeutic option because the response rate to liver-directed treatment is still around 70%. Sorafenib is a protein kinase inhbiter, including VEGFR, PDGFR, and RAF kinase. When combined with liver-directed therapy, sorafenib was found to show acceptable safety profiles and marginally improved efficacy. The great majority of patients required dose delays and/or reductions [137].

Less persuasive are the most recent phase 3 findings examining sorafenib’s advantages in the adjuvant situation following embolization [138]. Sorafenib has not been researched in the neoadjuvant situation before either liver-directed therapy, resection, or transplant. This adds potential periprocedural or postoperative problems that could jeopardize the successful administration of therapy, the life of the patient, or the graft, according to the lessons learnt from the use of various antiangiogenic substances in the neoadjuvanttherapy. For instance, with catheter-based procedures, arterial pruning brought on by antiangiogenic drugs may have an effect on how the small micron particle is delivered into the tumor bed.

There are a number of reasons to refrain from administering sorafenib prior to surgery when liver transplantation is involved. Due to nutritional deficiencies, transplant patients are frequently at a higher baseline risk of wound-healing complications; arterial complications are devastating and frequently fatal, and the nearly 70% stable disease rate seen in phase 3 trials may conceal the underlying metastatic disease, which could make a transplant inappropriate. In other cancer subtypes, antiangiogenic medications and radiation have been successfully combined. Therefore, research examining the combination of stereotactic body radiation therapy plus sorafenib or internal radiation (yttrium-90) plus sorafenib appear appropriate, and information on these combination regimens should become available over the next several years.

## 8. Conclusions and Challenges for the Future

HCC primarily develops from cirrhosis, and these two diseases cause more than two million fatalities annually around the world. Since liver cirrhosis and HCC have limited available treatments, it is crucial to stop the spread of these diseases as early as possible. In recent years, it has become evident that the gut microbiota, via the gut–liver axis, significantly contributes to the development of HCC. Dysbiosis can be caused by a variety of lifestyle variables, and it has been shown in numerous human trials and animal studies that short-term treatment with probiotics or synbiotics can reverse dysbiosis and consequently enhance liver health. Fatty liver disease is one instance where this is especially accurate. As a means to interfere with the effects of dysbiosis, TLR4 inhibitors are expected to enter clinical development. A possible method of lowering the occurrence of HCC is to prevent the early onset of progressive liver disease through the microbiome. Probiotics and synbiotics, as well as dietary changes, are possible methods for achieving this. As addressed above, the technological advances in liver cancer biomarker profiling have provided a breakthrough in liver metabolomics research.

In addition, challenges connected to HCC metabolism, cellular interfaces, metabolomics, and metabolic changes have been addressed. Future review research initiatives are proposed for improving HCC performances with clinical metabolomics:Our review indicates a unique liver cancer–metabolomics connection for therapeutic biomarker invention in HCC.Liver cancer remains one of the most difficult disease to treat; however, finding the therapeutic biomarker is possible.In single-cell studies of liver cancer, the phenomenon of extensive tumor heterogeneity has been noticed, which creates a major barrier for effective cancer interventions.Exploiting scientific systems to disrupt these interactions could establish a viable therapeutic strategy for targeting HCC and stopping HCC evolution, thereby improving treatment efficacy.We propose that clinical metabolomics may reflect the evolution of therapeutic biomarkers in a successful liver cancer treatment.

## Figures and Tables

**Figure 1 ijms-24-00537-f001:**
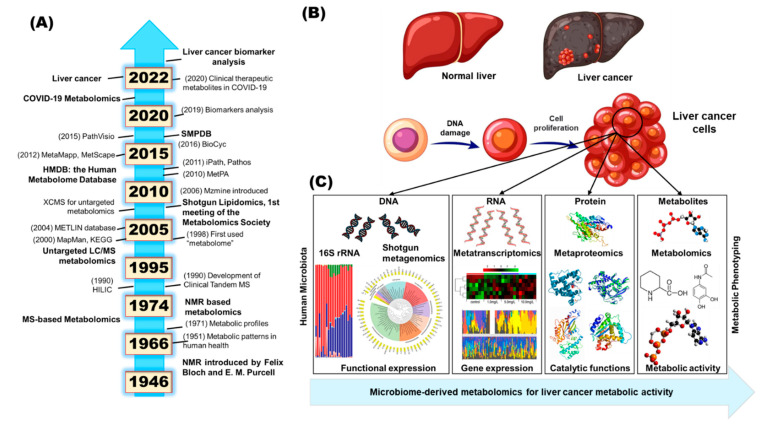
(**A**) The long history of metabolomics; timeline of major research and development milestones related to metabolomics and their medical applications. (**B**) Normal liver and liver cancer cells. (**C**) A schematic representation showing the multiomics cascade of systems biology. The multiomics analysis is influenced by epigenetics, toxicity, disease, and other environmental exposures. Here, metabolic communication within cells is carried out by DNA (metagenomics), RNA (metatranscriptomics), protein (metaproteomics), and metabolites (metabolomics).

**Figure 2 ijms-24-00537-f002:**
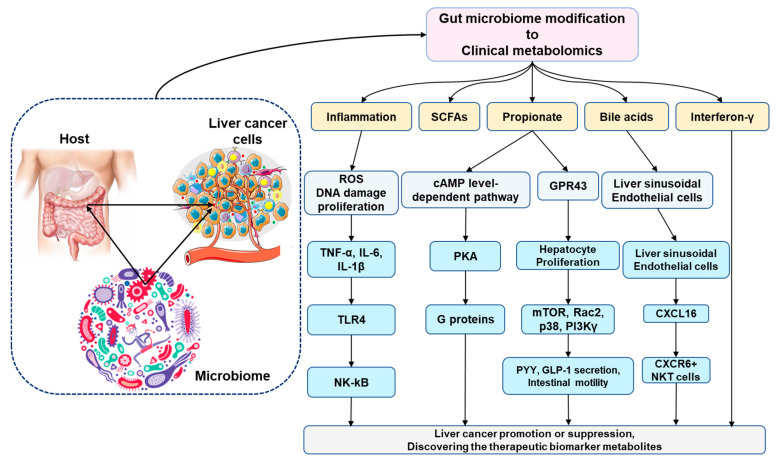
A three-way relationship between the host, tumor microenvironment, and microbiota. The gut microbiome-derived metabolomics target in HCC has been summarized. The host microbiome and metabolic reprogramming for cancer cells and their microenvironment may be related. An altering gut microbiota may boost propionate synthesis, which may reduce the risk of HCC both through a cAMP level-dependent mechanism and by interacting with GPR43. The gut microbiota alteration may result in an anti-HCC impact by boosting the amount of hepatic CXCR6+, NKT cells, and IFN-γ production. Primary-to-secondary bile acid conversion, which is controlled by the gut microbiota, influenced the CXCL16 expression of liver sinusoidal endothelial cells, which in turn affected the accumulation of CXCR6+ and NKT cells. cAMP, cyclic adenosine monophosphate; GPR, G protein-coupled receptor; IFN, interferon; IL, interleukin.

**Figure 3 ijms-24-00537-f003:**
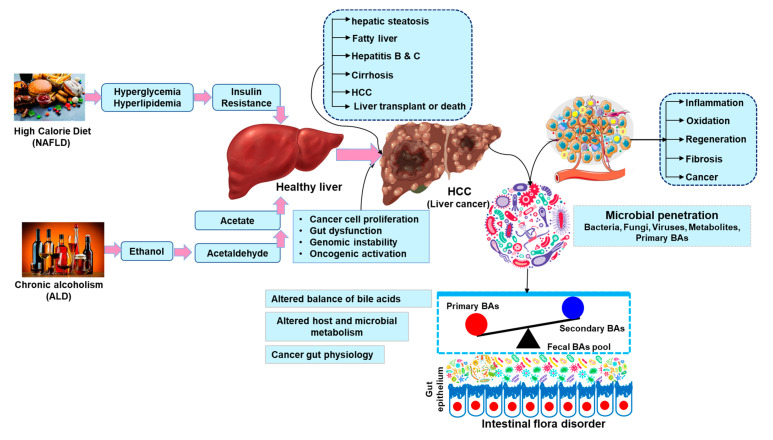
Molecular processes connected to both alcoholic and nonalcoholic HCC. The main risk factors for the development of NAFLD and AFLD, respectively, are a high-calorie diet and excessive alcohol intake. The clinical spectra of liver damage in supporting HCC development in NAFLD and AFLD have comparable molecular mechanisms despite the diverse pathogenic genesis. Within the intestinal tract, microbe-dependent reactions transform primary bile acids into secondary bile acids.

**Table 2 ijms-24-00537-t002:** Computational tools used for metabolomic technologies in biological samples.

Platforms	Invention	Ref
MetaboAnalyst	Web-based analytical pipeline tool, all-in-one metabolomics profiling, data collection, pathway enrichments, data analysis.	[24,54,55]
SIMCA-P+	Pattern recognition of PCA, PLS-DA, OPLS-DA, S-plot, and loading plot, multivariate tool, data mining, interactive graphics.	[11,13,56,57,58]
Chenomx Inc.,	Correction of the spectral data, metabolite profiling, and quantification.	[58,59,60]
MetExplore	Picturing of biological reaction systems and paths, simplifying the analysis of omics data in the biochemical background, and pathways improvement.	[61,62]
HMDB	Data bank of NMR, LC-MS, and GC-MS packs, metabolites information, structures, and biological properties.	[63,64,65]
KEGG	Databank of genes and genomes; KEGG ortholog for genes and proteins.	[66]
Reactome	Information base of biomolecular paths: free/open-source data, curated, and peer-reviewed.	[67,68]
Cyc databases	Largest curated collection of metabolic pathways. A wide range of model organisms’ data.	[69]
Virtual Metabolic Human	255 diseases, microbial genes, and human and gut microbiome metabolism database.	[70,71]
WikiPathways	Browsable, editable database curated by the research community.	[72]
Metabox	Toolbox for integrating proteomics and transcriptomics data for metabolomics data processing and interpretation.	[73]
Metscape	Cytoscape plugin, metabolomics correlation networks and KEGG-based metabolic networks integrating gene expression and metabolomics.	[74]
ChemRICH	Alternative to biochemical pathway mapping for metabolomic datasets. Not based on biochemistry directly but on structural similarity. The enrichment test is based on the Kolmogorov−Smirnov test (not the hypergeometric test or Fisher’s exact test).	[75]
PathBank	Comprehensive, user-friendly resource for metabolic pathways in 10 different model organisms.	[76]
OmicsNet	Multi-omics data integration, biological networks (genes, proteins, microRNAs, transcription factors, metabolites).	[77]
GEM-Vis	The use of metabolic network maps to visualize time-course metabolomic data.	[78]
FEMTO	Combining metabolomic time-series analysis with network data.	[79]

Notes and abbreviations: SIMCA, Soft Independent Modeling of Class Analogy; HMDB, Human Metabolome Database; KEGG, Kyoto Encyclopedia of Genes and Genomics; ChemRICH, Chemical Similarity Enrichment Analysis; GEM-Vis, Genome-Scale Metabolic Model Visualization; FEMTO, Functional Evaluation of Metabolic Time Series Observations; Ref, References.

**Table 3 ijms-24-00537-t003:** HCC-related changes in the microbiota’s regulation in animal, rat, and human models. Predominant microbiota present on various sites and their regulation with HCC.

Models	Disease	Implicated Microbiota	Ref
Mice	DEN-induced HCC	Changing gut microbiome	[111]
	DEN-CCL4-induced HCC	Changing gut microbiome	[112]
	STZ-HFD-induced NASH-HCC	*Atopobium* spp. *↑* *, Bacteroides* spp. *↑* *, Bacteroides vulgatus* *↑* *, B. acidifaciens* *↑* *, B. uniformis* *↑*, *Clostridium cocleatum* *↑* *, C. xylanolyticum* *↑* *, Desulfovibrio* spp. *↑*	[113]
	HFHC-induced NAFLD-HCC	*Mucispirillum**↑**, Desulfovibrio**↑**, Anaerotruncus**↑**, Desulfovibrionaceae**↑**, Bifidobacterium**↓*, *Bacteroides**↓*	[114]
	DMBA-HFD-induced HCC	Changing gut microbiome	[115]
	MYC transgenic spontaneous HCC	Gram-positive bacteria ↑, Bacteria mediating primary-to-secondary bile acid conversion ↑, *Clostridium scindens ↑*	[38]
	DMBA- or DMBA-HFD-induced HCC	Gram-positive bacteria	[116]
Rat	DEN-induced HCC	*Lactobacillus species**↓**, Escherichia coli**↑**, Atopobium cluster**↑**, Atopobium**↑**, Collinsella**↑**, Coriobacterium**↑**, Eggerthella**↑*, *Enterococcus species**↓**, Bifidobacterium species**↓**,*	[117]
Human	HCC	*Escherichia coli* *↑*	[118]
	HCC	*Cetobacterium* *↓* *, Proteobacteria* *↑* *, Desulfococcus* *↑* *, Enterobacter* *↑* *, Prevotella* *↑* *, Veillonella* *↑* *,*	[119]
	HCC	*Bifidobacterium* *↓* *, Bacteroides* *↑* *, Akkermansia* *↓* *,*	
	HCC	*Neisseria* *↑* *, Enterobacteriaceae* *↑* *, Veillonella* *↑* *, Limnobacter* *↑* *, Enterococcus* *↓* *, Phyllobacterium* *↓* *, Clostridium* *↓* *, Ruminococcus* *↓* *, Coprococcus* *↓*	[120]
	HCC	*Gut microbial α-diversity* *↓* *, Proteobacteria* *↑* *, Enterobacteriaceae* *↑* *, Bacteroides xylanisolvens* *↑* *, B. caecimuris* *↑* *, Ruminococcus gnavus* *↑* *, Clostridium bolteae* *↑* *, Veillonella parvula* *↑* *, Oscillospiraceae* *↓* *, Erysipelotrichaceae* *↓*	[121]
	HCC	*Klebsiella* *↑* *, Haemophilus* *↑* *, Alistipes* *↓* *, Phascolarctobacterium* *↓* *, Ruminococcus* *↓*	[122]

Notes and abbreviations: ↑, increased bacterial metabolism; ↓, decreased bacterial metabolism; HCC, hepatocellular carcinoma; DEN, Diethyl nitrosamine; HFD, high-fat diet; NASH, nonalcoholic steatohepatitis; Ref, Reference.

## Data Availability

Data are contained within the article.

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
