# Peer review of "Microbiome and Metabolomics in Liver Cancer: Scientific Technology"

_ijms, 2022, doi:10.3390/ijms24010537_

Round 1
Reviewer 1 Report
This review does not meet the expectations raised in the title: about half of it are off-topic, and the focus of Microbiome and Metabolomics is barely recognizable. Rather, this paper is a superficial description of a broad range of issues associated with liver cancer, that also includes matabolomic technologies and methods.
I cannot recommend publication, therefore.
Author Response
This review does not meet the expectations raised in the title: about half of it are off-topic, and the focus of Microbiome and Metabolomics is barely recognizable. Rather, this paper is a superficial description of a broad range of issues associated with liver cancer, that also includes matabolomic technologies and methods. I cannot recommend publication, therefore.
Response: We are grateful for the reviewer’s valuable comments. As per your concerns, we added the following details. In this review article, we carefully elaborate the clinical metabolomics in HCC. Finally, very important candidate scientific technologies in metabolomics (Table 1 and 2) and related potential metabolic pathways were discussed in this review points.
Main points in this revised manuscript.
Table 3 is newly added. Microbiome role in mice and human are summarized. Microbiome part is expanded.
Figure 1 is revised
Figure 2 is revised
Figure 3 is revised
In conclusion, take home message (5 points) of this review is added.
At this stage, we scientifically improved the manuscript to accept in journal.

Reviewer 2 Report
The authors have chosen as a review topic liver cancer and how changes in the microbiota can be used as an element of diagnosis, monitoring and therapy.
Hepatocellular carcinoma (HCC) is the main type of liver tumor and stands as one of the most common and deadly cancer worldwide. HCC is characterized by a complex molecular heterogeneity and carcinogenesis; the role played by the microbiota may be key in its progress and evolution. For this reason, research and knowledge of the role played by the microbiota is essential.
The authors have demonstrated great experience in the role played by the microbiota in different liver diseases, so the proposed review is correct but it would be necessary to make any changes
Minor Changes
The numbering of the different sections should be changed and the number 1 appears 3 times
1. Introduction (line 23)
1. Metabolomics (line 70)
1. Diagnostic tests of liver cancer (line 171)
Line 9: Change myriad by another world (a wide range/multitude)
Line 9: Change rewiring (is a term mainly used to talk about electricity) for a more correct term.
Line 72: The term ALD appears and has not been previously defined. Later they also appear on line 87 of the manuscript.
Table 1: It would be convenient to enlarge it, similar to table 2, and delimit the lines to ensure correlation in both but especially in table 2.
Line 252: Given the title chosen for the review the authors should increase the content of the section on Gut microbiota in HCC. Include if possible any patient or animal studies that have assessed specific changes in the microbiota in patients or animal models of HCC, indicating which microbes are overexpressed or depressed.
On the other hand, I find the figures made by the authors are fantastic and very illustrative.
Author Response
Comments and Suggestions for Authors
The authors have chosen as a review topic liver cancer and how changes in the microbiota can be used as an element of diagnosis, monitoring and therapy. Hepatocellular carcinoma (HCC) is the main type of liver tumor and stands as one of the most common and deadly cancer worldwide. HCC is characterized by a complex molecular heterogeneity and carcinogenesis; the role played by the microbiota may be key in its progress and evolution. For this reason, research and knowledge of the role played by the microbiota is essential.
Response: We agree to your comment. Much thanks for your wonderful scientific comments and appreciation of our review work. We significantly improved the way that microbiome roles in HCC are included and briefly expanded. The microbiome plays an important role as a metabolic modifier. This present revised manuscript has been covered with suitable explanations and literature data. Also, the conclusion part is revised.
The authors have demonstrated great experience in the role played by the microbiota in different liver diseases, so the proposed review is correct but it would be necessary to make any changes
Response: We value the reviewers’ comments. As per your scientific comments, we have revised the manuscript.
Minor Changes
The numbering of the different sections should be changed and the number 1 appears 3 times
- Introduction (line 23)
Response: The sentence is now corrected in the revised manuscript.
- Metabolomics (line 70)
Response: The sentence is now corrected in the revised manuscript.
- Diagnostic tests of liver cancer (line 171)
Response: The sentence is now corrected in the revised manuscript.
Line 9: Change myriad by another world (a wide range/multitude)
Response: we revised the sentence. Thank you.
Line 9: Change rewiring (is a term mainly used to talk about electricity) for a more correct term.
Response: we edited the sentence. Thank you.
Line 72: The term ALD appears and has not been previously defined. Later they also appear on line 87 of the manuscript.
Response: we explained the ALD (alcoholic liver disease). we are corrected in manuscript.
Table 1: It would be convenient to enlarge it, similar to table 2, and delimit the lines to ensure correlation in both but especially in table 2.
Response: As reviewer pointed out, we edited table 1.
Line 252: Given the title chosen for the review the authors should increase the content of the section on Gut microbiota in HCC. Include if possible any patient or animal studies that have assessed specific changes in the microbiota in patients or animal models of HCC, indicating which microbes are overexpressed or depressed.
Response: We agree with the reviewer’s comment. It is now corrected in the revised manuscript. We have now attached the Table 3 in the revised manuscript. It explained the gut microbiota in HCC from mice, rat and human studies.
On the other hand, I find the figures made by the authors are fantastic and very illustrative.
Response: Thank you for your comments. We made all figures to avoid the conflicts with others.

Reviewer 3 Report
The manuscript entitled "Microbiome and Metabolomics in Liver Cancer: Scientific Technology" by Raja Ganesan and Ki Tae Suk , it will be improved if the followings are addressed.
- Liver cancer can be caused by many different mechanism, e.g. viral, alcoholic, obesity.. etc. it will be nice if the authors elaborate more on the various metabolomic change/studies among various liver cancer types.
- Since the title of the article has the word "microbiome", I would expect more and I believe a table summarizing the names/identifies of good and bad microbiota that are present in the liver and/or surrounding environment.
- The data in the tables are a bit busy and should be better arranged using more columns and entering more useful information that are difficult to be described in the main text.
- For the names of the microbiota, please be consistent in style, i.e. Latin name should be italicized.
- At the end, any clear take home message to the audiences, e.g. the useful metabolomic/microbiota signature that can be used as early biomarker for liver cancer?
- Typos and unfriendly mode of English usage can be found.
Author Response
Comments and Suggestions for Authors
The manuscript entitled "Microbiome and Metabolomics in Liver Cancer: Scientific Technology" by Raja Ganesan and Ki Tae Suk, it will be improved if the followings are addressed.
- Liver cancer can be caused by many different mechanism, e.g. viral, alcoholic, obesity. etc. it will be nice if the authors elaborate more on the various metabolomic change/studies among various liver cancer types.
Response: We appreciate the reviewer's insightful criticism. Thank you very much for your scientific insights and support of our efforts. We have stated that microbiome may adversely improve metabolic remodeling, metabolites, and small molecule reactions. We directly concluded the concept that liver cancer would increase metabolic stress leading to metabolic scientific technologies to prevent liver cancer and death.
- Since the title of the article has the word "microbiome", I would expect more and I believe a table summarizing the names/identifies of good and bad microbiota that are present in the liver and/or surrounding environment.
Response: We have now attached the Table 3 in the revised manuscript with microbiota in HCC. We are deeply explained about microbiome in HCC.
- The data in the tables are a bit busy and should be better arranged using more columns and entering more useful information that are difficult to be described in the main text.
Response: We apologize for the unclear sentence. It is now corrected in the revised manuscript.
- For the names of the microbiota, please be consistent in style, i.e. Latin name should be italicized.
Response: We apologize for the unclear sentence. It is now corrected in the revised manuscript.
- At the end, any clear take home message to the audiences, e.g. the useful metabolomic/microbiota signature that can be used as early biomarker for liver cancer?
Response: We provided the take home message (5 points) at the end of conclusion part. the conclusion part is revised.
- Typos and unfriendly mode of English usage can be found.
Response: We did the official language edition (MDPI English editing ID: English-55734; manuscript ID: ijms-1967484). We are scientifically improved the manuscript. Thanks for your comments.

Round 2
Reviewer 1 Report
The title now appropriately reflects the contents and several other improvements were made.
Reviewer 3 Report
The authors have addressed to most of my concerns.